

# Fear of COVID-19 and illicit drug use during COVID-19 pandemic in Japan: a case-control study

Katsuya Nitta[1,*], Haruaki Naito[1,2,*], Takahiro Tabuchi[3] and Yasuhiro Kakiuchi[1,2]

[1] Department of Forensic Medicine, Faculty of Medicine, Kindai University, Osakasayama, Osaka, Japan
[2] Department of Forensic Medicine, Faculty of Medicine, Tokai University, Isehara, Kanagawa, Japan
[3] Cancer Control Center, Department of Cancer Epidemiology, Osaka International Cancer Institute, Chuo-ku, Osaka, Japan
[*] These authors contributed equally to this work.

## ABSTRACT

**Background**. Some people use illicit drugs to relieve stress. However, these drugs cause serious damage not only to individuals but also to society as a whole. Stress caused by the COVID-19 pandemic is considerable, as the number of illicit drug users continues to increase, despite a decrease in the availability of drugs and opportunities to leave the house during the pandemic. Fear of COVID-19 causes stress; however, its association with illicit drug use is not yet understood. In this study, we examined whether the fear of COVID-19 affects the subsequent use of illicit drugs.

**Methods**. We conducted a retrospective longitudinal case-control study using data from an Internet survey performed annually between 2020–2022, with the 2020 survey as the baseline survey and the 2021 and 2022 surveys as follow-up surveys. Those who were illicit drug use-free at the baseline survey but had a history of drug use upon follow-up were defined as the outcome group, whereas those who remained illicit drug use-free at follow-up were defined as the no-outcome group. Logistic regression analysis was conducted between the two groups, using "the fear of COVID-19" as the explanatory variable and adjusting for the effects of confounding factors. The same analysis was conducted by dividing illicit drugs into cannabis and non-cannabis groups, then setting their use as a secondary outcome.

**Results**. The study included 17,800 subjects, 837 of whom used illicit drugs at follow-up and 16,963 who did not use illicit drugs at follow-up. Logistic regression analysis revealed that higher levels of fear over COVID-19 correlated with higher illicit drug use among the participants. However, our analysis of cannabis-only outcomes showed no significant differences.

**Conclusions**. We found that fear of COVID-19 was a contributing factor to illicit drug use. Although the exact mechanism through which fear influences illicit drug use remains unknown, previous studies have shown that fear of certain targets increases illicit drug use, and our study adds to this evidence. However, in this study, we were unable to show a statistically significant causal relationship between fear of COVID-19 and the use of cannabis alone. Further research on the relationship between fear and the use of cannabis or other drugs, for varying focuses of fear, may broaden our knowledge of the different reasons individuals have for using different drugs.

Corresponding author
Yasuhiro Kakiuchi,
kakiuchi@med.kindai.ac.jp

## INTRODUCTION

The widespread use of illicit drugs is an important problem from the perspective of social safety and public health. In 2021, approximately 296 million people worldwide were using illicit drugs, or approximately 3.8% of the world population (*United Nations Office on Drugs and Crime, 2023*; *Statista, 2024b*). The definition of illicit drugs varies by country, but several drugs are highly addictive and can cause substance use disorders, with even small quantities posing serious health risks to humans. Although worldwide data has not been compiled, the number of drug overdose deaths in the United States, one of the countries with the highest number of illicit drug users, has been increasing every year, with more than 108,000 deaths reported in 2022 (*National Institute on Drug Abuse, 2023*).

The COVID-19 outbreak in Wuhan City, China, at the end of 2019 quickly spread worldwide and caused an increase in the number of deaths worldwide, thus severely impacting people's lives (*CDC, 2023*). During the COVID-19 pandemic, the global distribution of all substances was disrupted, and illicit drugs became more difficult to obtain (*Abdelnour et al., 2020*; *Conway et al., 2022*; *Butt & Alghababsheh, 2023*). Countries imposed various restrictions on outings, such as closing schools, restaurants, and bars; banning parties; and locking down cities, which reduced opportunities to buy illicit drugs outside or use them with more than one other person (*Americas Society/Council of the Americas, 2021*; *Velias, Georganas & Vandoros, 2022*; *Deimel et al., 2022*). Despite this, the number of illicit drug users has increased worldwide since the COVID-19 pandemic (*Statista, 2024a*). An increase in the number of cases of single-person use rather than multiple-person use has been observed, suggesting the existence of a strong craving for illegal drugs that outweighs the effort to obtain them, and a strong desire to use them (*Galarneau et al., 2021*). People may use illicit drugs in an attempt to relieve various mental health problems (*Sinha, 2008*). Given the current situation, the COVID-19 pandemic has placed considerable stress on people from a variety of perspectives, leading to an increase in the number of drug users (*Czeisler et al., 2020*; *Chacon et al., 2021*; *Conway et al., 2022*).

Historically, the fear of infectious diseases that pandemics instill in people has been found to lead to mental health problems, and similarly, the fear of COVID-19 has been reported to be associated with mental health problems (*Esterwood & Saeed, 2020*; *Koçak, Koçak & Younis, 2021*; *Chen et al., 2021*). However, to the best of our knowledge, no studies have yet examined the association between fear of COVID-19 and illicit drug use. Even broad investigations into fear utilizing statistical analytical methods are relatively scarce, underscoring this as a field with considerable potential for development. Previous studies have pointed to associations between illicit drug use and general fear (*e.g.*, fear of the living environment in certain neighborhoods) and diagnosed psychiatric disorders such as anxiety disorders and phobias (*Sareen et al., 2006*; *Smith & Book, 2008*; *Theall, Sterk & Elifson, 2009*). Fear is a varied subject, and further research may reveal differences in the association between illicit drug use and different types and degrees of fear. This study

represents the first global effort to investigate the link between fear of COVID-19 and subsequent illicit drug use.

## MATERIALS & METHODS

### Data collection

We obtained data from the Japan COVID-19 and Society Internet Survey (JACSIS). The JACSIS is an Internet-based, self-reported questionnaire survey conducted by Rakuten Insight, Inc., a leading Japanese Internet-based research agency (*Rakuten Insight Inc, 2002*). The JACSIS questionnaire consists of items related to the daily living, health, and economic activities of the population—including matters related to COVID-19. This survey is conducted once per year, between August to November, with the first one taking in August 2020 (*i.e.,* shortly after the start of the COVID-19 pandemic). The survey instrument was sent by the survey agency to panelists who were randomly selected based on the Japanese population distribution in the 2019 Comprehensive Survey of Living Conditions (CSLC), in order to avoid sample bias introduced by age, sex, and prefecture of residence. The panelists who agreed to participate in the survey accessed a designated website and completed the questionnaire. Participants could choose not to complete the survey and withdraw at any point. In this study, we conducted a retrospective longitudinal study using data from the 2020 JACSIS as the baseline and data from the 2021 or 2022 JACSISs as the follow-up data. The baseline survey was conducted from August 25, 2020 to September 30, 2020 (*The Japan COVID-19 and Society Internet Survey, 2023*). The follow-up surveys were conducted between September 27 and October 29 of 2021, and between September 12 and October 19 of 2022.

### Participants

Figure 1 presents the flowchart of participant selection. In the 2020 survey, the survey instrument was distributed online to 224,389 panelists. The respondents were randomly sampled so that their composition was representative of the Japanese population, yielding a sample of 28,000. The response rate was 12.5%. This is lower than the 37.9% Internet response rate for the Census (a survey administered by the Japanese government) but no definitive benchmark for response rates in online surveys has been defined. Some studies have suggested that whether the respondents are representative of the broader population is more significant than high response rates (*Ministry of Internal Affairs and Communications, 2021*; *Wu, Zhao & Fils-Aime, 2022*). Since the population of this study were all Japanese and the participants were sampled to fit the distribution of the overall Japanese population, the sample was deemed adequate despite the low response rate. To improve data quality, inconsistent or unnatural responses were considered invalid and excluded from the analysis. Specifically, respondents were considered invalid if they did not select the response that they were instructed to select in Q12 of the survey ($n = 1,955$) or if they answered that they had all diseases in Q75 ($n = 187$). The number of participants who gave incorrect responses and were excluded was 2,109, and the number of valid responses was 25,891. Additionally, participants with a history of illicit drug use at the baseline

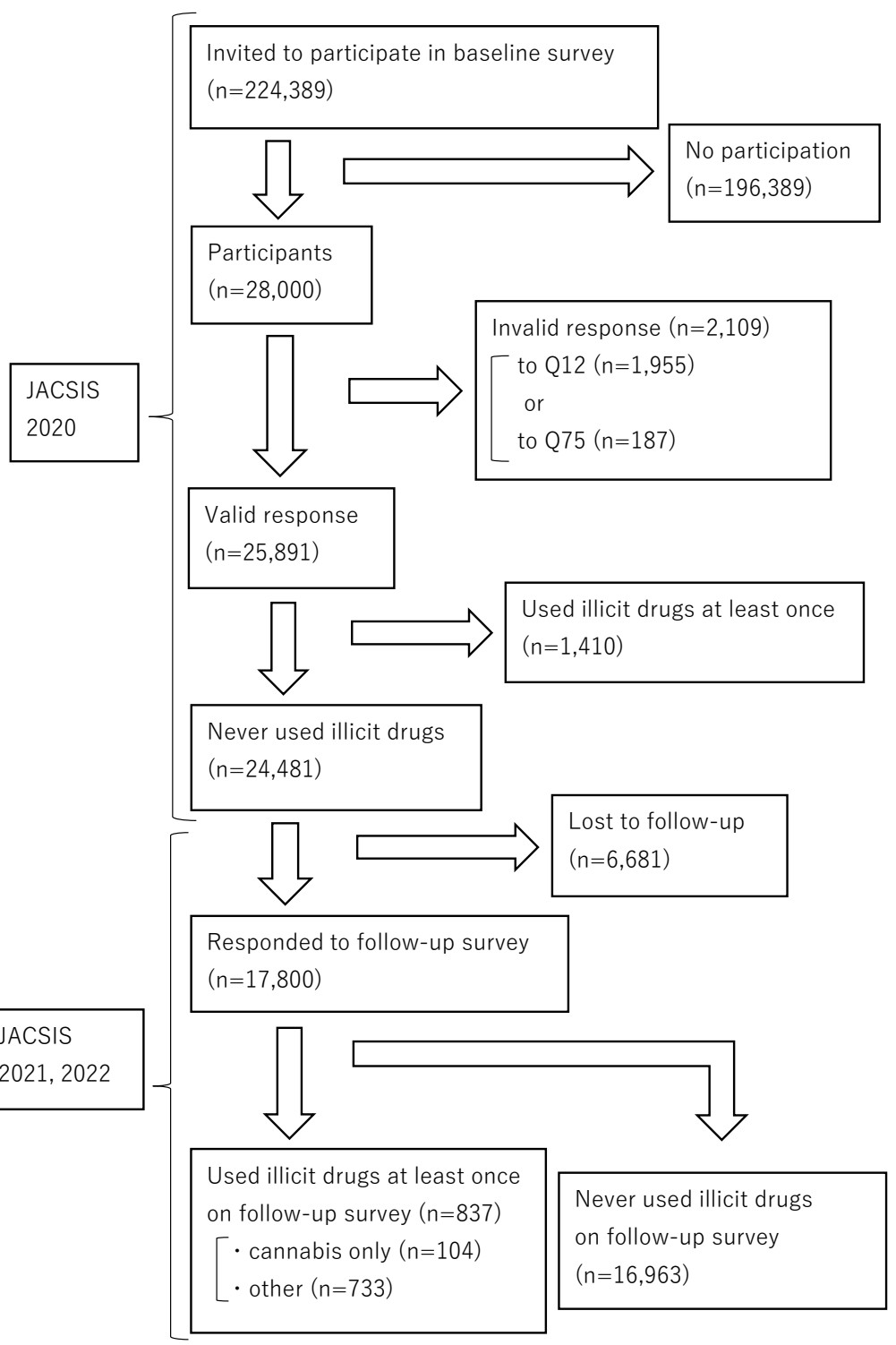

**Figure 1  Flowchart of study participants.** JACSIS, Japan COVID-19 and Society Internet Survey.

($n = 1,410$) and those who did not participate in the follow-up survey ($n = 6,681$) were excluded, leaving 17,800 participants in the final analysis.

## Outcome variables

The primary outcome measure was the occurrence of illicit drug use. Those who had used illicit drugs at least once at the time of the follow-up survey ($n = 837$) were included in the outcome group, and those who reported no history of illicit drug use at the follow-up ($n = 16,963$) were included in the no-outcome group. There are various types of illicit drugs, with varying effects on the human body (*Lundqvist, 2005*). Although overdoses of many of these can lead to death, there are currently no reports in the literature of cannabis overdose as a direct cause of death (*National Institute on Drug Abuse, 2019*; *CDC, 2024*). Moreover, although the use of cannabis for any purpose is illegal in Japan, some countries and states have legalized its recreational use—in contrast to other drugs (*Wilkinson et al., 2016*; *Volkow et al., 2016*; *Government of Canada, 2022*). Therefore, we categorized illicit drugs into two groups: cannabis and non-cannabis. Those who used only cannabis among the illicit drugs ($n = 104$) and those who used other drugs ($n = 733$) were considered the secondary outcome group. Because of the "gateway drug" aspect of cannabis and the inclusion of those who used stronger drugs after having used cannabis, the latter group was not asked whether they had used cannabis or not (*Secades-Villa et al., 2015*; *Balon, 2018*). The no-outcome group included patients who did not use any illicit drugs.

In the question about whether they had used illicit drugs, the answers "No (at least once but did not habitually)," "No (used habitually)," "Sometimes," and "Almost daily" were defined as "Yes, I have used," and "Never" was defined as "No history of use." Illicit drugs were defined as substances such as morphine and other narcotics obtained without a prescription from a physician; organic solvents such as thinner and toluene; designer drugs such as law-evading herbs and magic mushrooms; cannabis; methamphetamine; cocaine; and heroin.

## Explanatory variable

We used the Fear of COVID-19 Scale (FCV-19S) as an explanatory variable. The Fear of COVID-19 Scale is a short questionnaire developed to assess fear and anxiety associated with COVID-19 (*Midorikawa et al., 2021*; *Ahorsu et al., 2022*). FCV-19S comprises seven questions in total: "I am very afraid of the novel coronavirus," "I get uncomfortable when I think about the novel coronavirus," "My hands sweat when I think about the novel coronavirus," "I am afraid of losing my life due to the novel coronavirus," "I get nervous and anxious when I see news and topics about the novel coronavirus on the Internet," "I cannot sleep because I worry about COVID-19," and "My heart beats faster and I have heart palpitations when I think about the COVID-19." Each question was rated on a 5-point Likert scale (1 = strongly disagree to 5 = strongly agree). The total score, which ranged from 7 to 35, was calculated by summing the scores for each question. A higher score indicates a stronger fear of COVID-19. To the best of our knowledge, no studies have clearly defined the severity of fear of COVID-19 using the FCV-19S and there is no cutoff score. Therefore, we divided the severity of fear into four levels, as in a previous study

(none, <17 points; mild, 17–22 points; moderate, 23–28 points; and severe, 29–35 points) (*Yoshida et al., 2023*).

## Statistical analysis

We used a multivariate binary logistic regression model to analyze the association between the level of fear of COVID-19 in the early stages of the COVID-19 pandemic and the presence of illicit drug use. We adjusted for potential imbalances in the following key associated factors between groups with and without outcome: sex (male, female), age (15–24, 25–34, 35–44, 45–54, 55–64, 65+), occupational status (employed, student, unemployed), number of persons in household (one, two or more), annual household income (>7 million yen, 3–7 million yen, <3 million yen, do not know/do not want to answer), marital status (married, unmarried), education (ISCED ≥6, ISCED<6, others), drinking status (never, rarely, sometimes, almost daily), smoking status (never, used to smoke but quit, current smoker), Kessler Psychological Distress Scale (K6) (negative, positive), and presence of any psychiatric disorder (no, yes).

The K6 is a 6-item questionnaire that assesses stress in the past month (*Kessler et al., 2002*). Respondents answered each question on a 5-point scale (0 to 4), and the total score was used to evaluate their level of stress. Total scores ranged from 0 (low stress) to 24 (high stress). In this study, a score of 5 or more, representing moderate psychological stress, was defined as stressed (*Sakurai et al., 2011*).

To verify the uniformity of the impact of fear on drug use, subgroup analyses were conducted *post-hoc*. Subgroups were defined according to smoking history (yes or no) and drinking status (current or nondrinker). The statistical significance of differences between subgroups was tested using interaction terms. No adjustments were made for multiple tests.

The variance inflation factor (VIF) was used to check for multicollinearity, with a significance level of 5%. SAS software (version 9.4, SAS Institute Inc., Cary, NC, USA) was used for analysis.

## Ethics approval

All procedures were conducted in accordance with the ethical standards of the 1975 Declaration of Helsinki. The research protocol was reviewed and approved by the Research Ethics Committee of the Osaka International Cancer Institute (approved June 19, 2020, approval number 20084). The Internet survey agency respects Japanese privacy laws. All participants provided electronic informed consent by reviewing and agreeing to a web-based instructional document detailing the survey prior to completing the online questionnaire. As an incentive, participants were given "E-points," credit points that could be used for Internet shopping or redeemed for cash. The precise value of "E-points" was not disclosed by the company that administered the Internet survey (*Sato et al., 2022*).

# RESULTS

## Participants characteristics

Table S1 presents the participant characteristics. The total number of participants included in this analysis was 17,800, with 837 individuals in the primary outcome group of illicit drug

users and 16,963 in the comparison group of non-users. In the secondary outcome group, 104 individuals used only cannabis and 733 used drugs other than cannabis (regardless of cannabis use).

The proportion of men was higher in all of the drug-use groups, while that of women was higher in the non-use group. The drug-use group had a higher percentage of younger individuals, and the most represented age group in the non-use group was those aged ≥65 years (29.1%). In terms of education, the drug-use group had a higher proportion of individuals with ISCED scores of ≥6, whereas the non-use group had more individuals with ISCED scores of <6. Regarding drinking status, the group that used only cannabis had a lower percentage of non-drinkers (10.6%), by ∼50% *vs* the other groups. Regarding smoking status, the group that used only cannabis had the highest proportion of ex-smokers (46.2%), whereas non-smokers comprised the highest proportion in the other groups. Occupational status, number of persons in the household, household income, marital status, K6 score, and the presence of any psychiatric disorder showed similar distributions across all of the groups.

### Primary & secondary outcomes

The results of the logistic regression analysis are presented in Table S2. The distribution of fear levels regarding COVID-19 was also similar across all of the groups, with the highest proportion being "mild" (∼40%), followed by "none", "moderate", and "severe."

In the group that used illicit drug, using "none" as the reference for the degree of fear of COVID-19, the adjusted odds ratios were 1.32 for "mild", 1.42 for "moderate", and 1.74 for "severe"—thus showing an increase with rising levels of fear. In the group that used only cannabis, no significant difference was observed in the number of cannabis users, owing to varying levels of fear. In the group that used drugs other than cannabis, the adjusted odds ratios were 1.35 for mild fear, 1.51 for moderate fear, and 1.86 for severe fear. Similarly to the primary outcome, this indicated that the odds ratios increased as the level of fear increased.

### Subgroup analysis

No significant interactions were observed between the subgroups in terms of drug use, across all of the groups (Fig. S1). In the groups that used illicit drugs and those that used drugs other than cannabis, consistent odds ratios for drug use were observed across all of the subgroups. In the group that used cannabis only, no statistically significant differences were found, and the results were inconsistent.

## DISCUSSION

We tested the hypothesis that fear of COVID-19 in the early stages of the COVID-19 pandemic would be associated with subsequent illicit drug use. Although several studies on fear of COVID-19 have been conducted, to the best of our knowledge, this is the first study on the association between illicit drug use and fear of COVID-19 (*Chacon et al., 2021*; *New York University Web Communications, 2022*; *Meller et al., 2022*).

Logistic regression analysis revealed that the higher the fear of COVID-19 in the early stages of the COVID-19 pandemic, the more likely the respondents were to use illicit

drugs later. When conducting multivariate analysis, confounding factors such as smoking, alcohol consumption, and mental health status were set as adjustment factors to minimize bias in the results.

In this study, we not only analyzed the outcome of overall illicit drug use, but also the secondary outcomes in the groups that used only cannabis and those that used drugs other than cannabis. Our results showed that there were no statistically significant differences in the outcomes according to level of fear over COVID-19 in the cannabis-only group. However, in the group that used drugs other than cannabis, as well as the group that used illicit drugs overall, the rate of illicit drug use increased with fear level. In other words, these results suggest that fear of COVID-19 may represent a cause of illicit drug use, but likely has no effect on the use of cannabis alone. Although previous studies have shown that fear of a local living environment within a certain neighborhood, anxiety disorders, and psychiatric disorders such as phobia represent significant causes of illicit drug use, the relationship between fear exposure and cannabis use alone is unknown because these studies only analyzed illicit drugs overall, rather than by drug type (*Sareen et al., 2006*; *Smith & Book, 2008*; *Theall, Sterk & Elifson, 2009*). In some users, cannabis can function as a "gateway drug" that promotes the use of harder illicit drugs. Thus, certain differences may exist in terms of phases of drug use (*Secades-Villa et al., 2015*; *Balon, 2018*; *Williams, 2020*). In addition, 42% of Japanese cannabis users perceive the drug as completely risk-free, which is considerably higher than the comparable figure of 2% for methamphetamine (*Organized Crime Department & National Police Agency in Japan, 2023*). Thus, it can be assumed that cannabis has a strong recreational aspect, and that people tend to use stronger-acting drugs to cope with more serious forms of psychological distress. Further analysis of other objects of fear may provide new insights into which focuses of fear are likely to increase the use of which illicit drugs.

However, the sample size of the cannabis-only use group in the current study was small ($n = 104$); therefore, it is possible that an existing significant difference simply could not be confirmed. Japan has a relatively small number of drug users; therefore, the sample size that can be collected is limited (*Health Canada, 2018*; *National Center for Drug Abuse Statistics, 2020*; *The Department of Drug Dependence Research, 2024*). Subgroup analyses revealed no significant differences regarding interactions. Although sex, smoking, and alcohol consumption represent factors that influence illicit drug use, these were not found to interact with fear of COVID-19—suggesting that fear of COVID-19 is a factor in illicit drug use that is independent of these others (*Yi et al., 2017*).

The baseline survey began in August 2020, less than one year after the COVID-19 pandemic struck the world (*CDC, 2023*). While various social policies were implemented in each country to cope with the pandemic, Japan declared a state of emergency to prevent people from leaving their household (*The Asia Pacific Journal, 2022*). At the time, the vaccine was still in development and was not yet available for sale; and with the daily news reports on the increasing number of infections and deaths, it was the time when fear of an unknown virus was high (*Chakraborty, Bhattacharya & Dhama, 2023*). However, the information available worldwide is not always correct. While the media are full of high-impact reports on the number of infected people and deaths, there are also programs

that provide expert commentary on viruses and infectious diseases. The problem lies in the amount of misinformation. With the advent of increased use of social media, it is easy for a single individual's assertion, which may or may not be correct, to be spread easily. Even during the COVID-19 pandemic, misinformation was widespread, and it was very difficult to discern correct information (*Ferreira Caceres et al., 2022*). Any information must be verified before being accepted as true. Obtaining correct information can help alleviate fear. Given the results of this study, the amount of illicit drug use could have been reduced if correct information regarding COVID-19 had been shared with more people, thus alleviating their fears (*Finset et al., 2020*; *Berhe et al., 2022*). Similar global pandemics and natural disasters are likely to occur in the future. Thus, it is important to disseminate correct information and knowledge, as well as properly address peoples' fears, so as not to further exacerbate the issue of illicit drug use, which is already significant even under regular circumstances.

The subject of fear can vary significantly, so it would also be interesting to broaden the scope of this study to include fears regarding personal and familial health, economic realities, and societal-level disasters such as pandemics.

This study has several limitations. First, a questionnaire was administered to collect data. Questionnaires are one of the most widely used tools for data collection because of their convenience; however, one should be aware that humans tend to choose socially desirable answers to sensitive questions (*Holtgraves, 2004*; *Krumpal, 2013*; *Suerken et al., 2014*; *Taherdoost, 2016*). This study dealt with the sensitive question of illicit drug use, and it is possible that some respondents falsely answered that they did not use illicit drugs even though they actually did. Second, the questionnaire in this study included items on the frequency of drug use but not on the amount of drug use; therefore, it was not possible to examine the severity of drug use. Future studies should include more questions on usage and other illicit drugs, which would allow for more research on severity and drug-specific perspectives.

## CONCLUSIONS

Fear of COVID-19 in the early stages of the COVID-19 pandemic has been found to be a cause of illicit drug use during the later stages. In the event of a pandemic caused by a new pathogen, disseminating the correct information and removing people's fears may be critical in preventing the spread of illicit drug use.

## ACKNOWLEDGEMENTS

We would like to thank Honyaku Center Inc. for English language editing.

### Funding

This research was supported by the Japan Society for the Promotion of Science (JSPS) KAKENHI (grant number 17H03589, 19K10671, 19K10446, 18H03107, 18H03062,

20H00040, and 21H04856), the JSPS Grant-in-Aid for Young Scientists (grant number 19K19439), the Research Support Program to Apply the Wisdom of the University to Tackle COVID-19 Related Emergency Problems, University of Tsukuba, the Health Labor Sciences Research (grant number 19FA1005, 19FA1012, 19FG2001, and 22FA2001), grants from Chiba Foundation for Health Promotion and Disease Prevention, Innovative Research Program on Suicide Countermeasures (R3-2-2), the Ministry of Health, Labor, and Welfare (MHLW) Special Research Program (grant number JPMH20CA2046), the Japan Agency for Medical Research and Development (AMED) (grant number 2033648), and JST RISTEX (grant number JPMJRX21K6). The funders had no role in study design, data collection and analysis, decision to publish, or preparation of the manuscript.

## Grant Disclosures

The following grant information was disclosed by the authors:
Japan Society for the Promotion of Science (JSPS) KAKENHI: 17H03589, 19K10671, 19K10446, 18H03107, 18H03062, 20H00040, 21H04856.
JSPS Grant-in-Aid for Young Scientists: 19K19439.
Research Support Program to Apply the Wisdom of the University to Tackle COVID-19 Related Emergency Problems.
University of Tsukuba.
Health Labor Sciences Research: 19FA1005, 19FA1012, 19FG2001, 22FA2001.
Grants from Chiba Foundation for Health Promotion and Disease Prevention, Innovative Research Program on Suicide Countermeasures: R3-2-2.
The Ministry of Health, Labor, and Welfare (MHLW) Special Research Program: JPMH20CA2046.
Japan Agency for Medical Research and Development (AMED): 2033648.
JST RISTEX: JPMJRX21K6.

## Competing Interests

The authors declare there are no competing interests.

## Author Contributions

- Katsuya Nitta analyzed the data, prepared figures and/or tables, authored or reviewed drafts of the article, and approved the final draft.
- Haruaki Naito analyzed the data, prepared figures and/or tables, authored or reviewed drafts of the article, and approved the final draft.
- Takahiro Tabuchi conceived and designed the experiments, performed the experiments, authored or reviewed drafts of the article, and approved the final draft.
- Yasuhiro Kakiuchi conceived and designed the experiments, analyzed the data, authored or reviewed drafts of the article, and approved the final draft.

## Human Ethics

The following information was supplied relating to ethical approvals (i.e., approving body and any reference numbers):

The research was approved by the Research Ethics Committee of the Osaka International Cancer Institute, approval number 20084.

## Data Availability

The raw data is available in the Supplementary File.

## Supplemental Information

Supplemental information for this article can be found online at http://dx.doi.org/10.7717/peerj.18137#supplemental-information.

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
