# Peer review of "Fear of COVID-19 and illicit drug use during COVID-19 pandemic in Japan: a case-control study"

_PeerJ, doi:10.7717/peerj.18137_

## Round 0.1 · original submission · Major Revisions

Following the review of Research Article titled "Fear of COVID-19 and illicit drug use during COVID-19 pandemic in Japan: a longitudinal study" I recommend that it should be revised taking into account the changes requested by the reviewers.

Reviewer 1 ·

Basic reporting

The paper is largely clear. A few suggestions were made to improve clarity in comments throughout the pdf.
The background on the impact of the pandemic on drug use is sufficient.
I understand that the authors are limited in drawing upon previous research on the impact of fear of COVID and drug use as we have not had a pandemic of this scale in over 100 years.
However, there is likely some literature on the impact of either individual-level fears (e.g. health concerns, financial stressors, etc.) or community-level fears (e.g. natural disasters) that could help inform this discussion.
The raw data was shared and the tables/figures are adequate.

Experimental design

no comment

Validity of the findings

The findings appear to be largely supported by the research. However, I believe that conclusions should be tempered to some extent. I do not believe the researchers can state with certainty that COVID fear accounts for all of the increased drug use seen in the sample. There may be other factors, either captured in the survey or not, that also explain some of the variance in drug use reported in the sample.

Additional comments

The authors mention damage to the environment in the abstract and introduction but this is not discussed or supported by previous literature. The environmental impact of drug use could be significant but this paper may not be the best place to address it.

I am not familiar with the in-text citation style used.

The response rate was relatively low so I think this merits some discussion. Is this rate common/acceptable for these types of surveys and in Japan?

I would be interested to see if the researchers could parse out any difference in the drugs being used. For example, if individuals started using cannabis more frequently, this may not be an urgent health concern. However, if they started using drugs that could contain fentanyl and carry an immediate risk of overdose death, this would be much more concerning.

I also wonder if the authors could expand their analysis/discussion of factors that are more strongly related to increased use as this could have intervention implications. For example, is the relationship between COVID fear and drug use stronger among smokers than non-smokers?

Annotated reviews are not available for download in order to protect the identity of reviewers who chose to remain anonymous.

·

Basic reporting

Thank you for submitting your article to PeerJ for potential publication. After reviewing your manuscript, I believe it holds promise due to its engaging topic. However, several areas require substantial enhancement to meet the publication standards of a full-length scholarly paper. Below, I offer detailed feedback aimed at guiding these improvements:

Experimental design

Introduction Revision: Currently, the introduction briefly mentions various topics, such as environmental degradation, without further exploration or integration into the main subject. This approach disrupts the logical flow and cohesion expected in scholarly writing. I recommend a thorough revision to ensure a more structured and focused introduction that directly relates to your study's core objectives. Enhancing the readability and flow of your paragraphs will also contribute to a stronger foundation for your paper. Methods Detailing: The manuscript lacks specific psychometric details concerning the instruments used in your research. Additionally, a more detailed quantitative description of your methods is necessary. Providing these details will not only bolster the credibility of your study but also allow for replication and verification of your results by other researchers.

Validity of the findings

Results Enhancement: The Results section is noticeably lacking in figures, numbers, and a detailed analysis of your findings. Reference to 'Picture 1' and subsequent results are either absent or only mentioned superficially. A more thorough presentation and discussion of your data are crucial for the reader to understand the significance and implications of your findings. Incorporating detailed figures and an in-depth analysis will significantly improve the quality of this section. Discussion Depth: The discussion appears superficial, particularly in its treatment of related literature on the fear of COVID-19. This section would benefit greatly from a more comprehensive review and discussion of existing studies, highlighting how your work contributes new insights or complements the current body of research. Discussing a broader range of literature will also help situate your study within the larger academic discourse. In summary, while your manuscript presents an interesting topic, it requires significant revisions to elevate its scholarly contribution. Addressing the specific areas mentioned above will not only clarify your research findings but also enhance the overall impact of your work. I look forward to seeing the improvements made to your manuscript.

---

## Round 0.2 · accepted · Accept

The authors have addressed all of the reviewers' comments.

·

Basic reporting

The revised version has been significantly improved. The introduction now serves the research topics, their aims, and objectives. Readability has been enhanced and the flow is much more compelling.

Experimental design

The design has been strengthened and the analyses have been enriched.

Validity of the findings

Findings are valuable and well-discussed.